# Effect of Hesperidin on Cardiovascular Disease Risk Factors: The Role of Intestinal Microbiota on Hesperidin Bioavailability

**DOI:** 10.3390/nu12051488

**Published:** 2020-05-20

**Authors:** Anna Mas-Capdevila, Joan Teichenne, Cristina Domenech-Coca, Antoni Caimari, Josep M Del Bas, Xavier Escoté, Anna Crescenti

**Affiliations:** 1Eurecat, Technology Centre of Catalunya, Nutrition and Health Unit, 43204 Reus, Spain; anna.mas@eurecat.org (A.M.-C.); joan.teichenne@eurecat.org (J.T.); cristina.domenech@eurecat.org (C.D.-C.); antoni.caimari@eurecat.org (A.C.); josep.delbas@eurecat.org (J.M.D.B.); 2Department of Biochemistry and Biotechnology, Universitat Rovira i Virgili, Campus Sescelades, 43007 Tarragona, Spain; 3Eurecat, Technology Centre of Catalunya, Biotechnology Area and Technological Unit of Nutrition and Health, 43204 Reus, Spain

**Keywords:** cardiovascular diseases, hesperidin, hesperetin, gut microbiota, dysbiosis, bioavailability

## Abstract

Recently, hesperidin, a flavonone mainly present in citrus fruits, has emerged as a new potential therapeutic agent able to modulate several cardiovascular diseases (CVDs) risk factors. Animal and in vitro studies demonstrate beneficial effects of hesperidin and its derived compounds on CVD risk factors. Thus, hesperidin has shown glucose-lowering and anti-inflammatory properties in diabetic models, dyslipidemia-, atherosclerosis-, and obesity-preventing effects in CVDs and obese models, and antihypertensive and antioxidant effects in hypertensive models. However, there is still controversy about whether hesperidin could contribute to ameliorate glucose homeostasis, lipid profile, adiposity, and blood pressure in humans, as evidenced by several clinical trials reporting no effects of treatments with this flavanone or with orange juice on these cardiovascular parameters. In this review, we focus on hesperidin’s beneficial effects on CVD risk factors, paying special attention to the high interindividual variability in response to hesperidin-based acute and chronic interventions, which can be partly attributed to differences in gut microbiota. Based on the current evidence, we suggest that some of hesperidin’s contradictory effects in human trials are partly due to the interindividual hesperidin variability in its bioavailability, which in turn is highly dependent on the α-rhamnosidase activity and gut microbiota composition.

## 1. Introduction

Cardiovascular diseases (CVDs) are the first cause of death in the world, causing about 31% of all deaths worldwide [1]. The development of CVDs are usually associated with the presence of several risk factors, some of them related with poor health habits [2,3,4]. Diet is a major external factor for CVDs development, and recommendations for the improvement of dietary and lifestyle routines and making them affordable and available for the general population are the primary approach to the prevention of the onset of this pathology [5,6]. In this sense, differences between dietary patterns, such as Mediterranean, Portfolio, Nordic, and vegetarian diet, are associated with different cardio-metabolic outcomes [7]. Nowadays, common therapies based on drugs are administered to patients who have already been diagnosed with any cardiovascular disorder [7]. Taking into account all the beneficial effects associated with the diet in CVDs development, the use of alternative treatments, such as natural-based products has gained importance as a preventive strategy for improving some CVD factors, such as hypertension, diabetes, cholesterol, and obesity [8]. In this sense, nowadays, research in food bioactive compounds is in the spotlight to develop new functional foods and nutraceuticals aimed at preventing and/or ameliorating CVD risk factors. 

Polyphenols are a large group of bioactive plant compounds for which beneficial effects in the prevention and treatment of different pathologies, including CVDs, have been described [9]. The main classes of polyphenols are flavonoids, which include flavanols (e.g., tea), flavanones (e.g., citrus fruits), and flavonols (e.g., tea, apples, and onions), and nonflavonoids, including hydroxycinnamic acids (e.g., coffee) and anthocyanins (e.g., cherry) [10]. Of them all, flavanones, including hesperidin and naringin as the principal molecules, are considered the main subclass of polyphenols [11]. Orange and its fruit juice, which are considered rich sources of hesperidin and naringin, are the most common citrus fruit products consumed among the European population [12,13]. Hesperidin and, far behind, naringin, represent more than 90% of the flavonoids in sweet oranges [14,15]. The highest concentrations of hesperidin are found in the solid tissues of citrus fruits, although considerable amounts are also found in their juices [16]. In the form of supplements or nutritional complements, hesperidin is considered innocuous, with limited adverse effects, due to its nonaccumulative nature [17]. 

Hesperidin and its derived intestinal metabolites are shown in Figure 1. The number and specific position of hydroxyl groups in the flavanones aromatic rings, which produce important changes in their biochemical structure, are considered to be crucial for the reported beneficial effects of citrus polyphenols [18,19]. Some of these effects include antitumor, antioxidant, anti-inflammatory, hypocholesterolemic, and hypoglycemic effects, related to an improvement in different pathologies, such as cancer, neurodegenerative diseases or CVDs [20,21]. Regarding CVDs, the results of in vitro and in vivo studies have shown that hesperidin treatment produces beneficial effects on different risk factors, including the improvement of lipid and glucose metabolism, adiposity, and hypertension [22,23,24,25,26]. However, the results of human studies aiming to decipher possible beneficial effects of hesperidin on CVD risk factors are inconclusive, as there are studies that do not demonstrate beneficial effects of hesperidin on such factors.

The molecular structure of hesperidin also affects its bioavailability and absorption levels [27]. Thus, the metabolism of citrus flavanones is determined by the sugar moieties and its removal degree by intestinal bacteria. Citrus flavanones are resistant to stomach and small intestine enzymes and thus reach the colon intact. There, intestinal microbiota activity breaks down the hesperidin molecule, releasing the aglycone form, named hesperetin [28,29] (Figure 1). Once inside the intestinal epithelium, hesperetin is released into the bloodstream in form of glucuronide and sulfatate conjugates [30]. In addition, an important part of the metabolized hesperetin is transformed by the microbiota present in the colon, generating some bioavailable and highly specific catabolites of hesperetin [30,31]. 

Bioavailability of hesperidin and its ability to produce beneficial effects can be modulated by different factors. Some of them are related with the food matrix and the physical form in which they are ingested (e.g., juice, soluble extract or capsules, among others), processing methods and storage techniques, as well as the structure of the compound and the host intrinsic characteristics, including intestinal microbiota composition [18]. All these factors affect the solubility of flavanones and their uptake by the gastrointestinal tract [28,29].

Considering that the intestinal microbiota activity plays an important role in the bioavailability of hesperidin and, as a consequence, its beneficial effects on CVD risk factors, it is tempting to shed light on its intrinsic mechanisms [14,31,32]. Therefore, the present review aimed to describe the effects of hesperidin consumption on CVD factors and to highlight hesperidin interindividual variability in its bioavailability and effectiveness, a process in which the gut microbiota plays an important role. This reported variability would explain the discrepancies observed between animal studies and human studies on beneficial effects of hesperidin over CVD risk factors.

## 2. Beneficial Effects of Hesperidin on Cardiovascular Disease Risk Factors

### 2.1. Effects of Hesperidin on Glucose Homeostasis

Diabetes is one of the major risk factors for developing CVDs. The main complication of diabetes is CVDs, and it is estimated that 65% of diabetic patients die from CVD complications [33]. In this sense, several studies have shown beneficial effects of hesperidin in glucose metabolism at the preclinical level, both in animal and in vitro models. 

At the in vitro level, neohesperidin (derived from hesperidin) treatment was shown to increase glucose consumption in the hepatocyte cell line HepG2, which was associated with increased phosphorylation levels of adenosine monophosphate (AMP)-activated protein kinase (AMPK) [34]. Xuguang et al. recently reported attenuated glucose content in culture medium and increased glucose uptake in lipopolysaccharide (LPS)-induced insulin-resistant HepG2 cells treated with hesperidin. These changes seemed to be associated with the regulation of the insulin receptor substrate 1 (IRS1)- glucose transporter (GLUT)-2 pathway via toll-like receptor (TLR)-4 [23]. This positive effect over glucose uptake was corroborated in another recent study, showing that both hesperidin and hesperetin exert antidiabetic properties in L6 myotubes by inducing glucose uptake and reducing oxidative stress and advanced glycation end-products (AGEs) formation [22]. Related to AGE formation, Irshad and collaborators recently showed that a combination of trans-resveratrol and hesperetin is able to dampen the rise of methylglyoxal levels caused by high glucose concentrations by increasing the expression of Glyoxalase (Glo)-1 and decreasing the expression of hexokinase (HK)-2 in human aortal endothelial cells [35]. 

There is accumulating evidence demonstrating the glucose-lowering effects and the improvement in insulin resistance parameters exerted by hesperidin both in Type-1 diabetes (T1D) [36,37,38,39,40,41] and Type-2 diabetes (T2D) [42,43,44,45,46] rodent experimental models, thus demonstrating the antidiabetic properties of hesperidin. These effects were shown to be achieved by the modulation of key glucose regulation enzymes, such as an upregulation of glucokinase (involved in glycolysis) or a downregulation of the gluconeogenic enzyme glucose-6-phosphatase [36,40,42,43,44]. Other effects of hesperidin treatment in diabetic animals include a reduction in inflammatory parameters, such as tumor necrosis factor alpha (TNFα), interleukin (IL)-6 or IL-1β, and the reduction of oxidative stress associated with diabetes [38,39,45]. Akiyama and collaborators also reported a recovery of adiponectin levels mediated by hesperidin both in T1D and T2D models [36,44]. In addition to the effects observed in diabetic models, improvement of glucose metabolism and insulin resistance were also described by our group and others in other animal models of human diseases that are associated with alterations in glucose metabolism, such as metabolic syndrome (MetS) and obesity [47,48,49,50,51].

Despite all evidence at the preclinical level, the effects of hesperidin consumption on glucose metabolism in humans are not conclusive. In a recent randomized, double-blind, placebo-controlled clinical trial, Yari et al. reported that daily consumption of hesperidin capsules (500 mg) for 12 weeks significantly decreased fasting glucose levels, both compared with basal levels and with placebo group in patients with MetS [52]. Decreases in insulin levels and in the homeostatic model assessment for insulin resistance (HOMA-IR) index were also reported, although no significant differences vs. the placebo group were observed in these parameters [52]. Ribeiro et al. reported a decrease of 18% in insulin levels and a reduction of 33% in HOMA-IR index after 12 weeks of daily consumption of 500 mL of orange juice (OJ) in obese individuals compared to control group [53]. Lima and collaborators also reported significant decreases in blood glucose and insulin fasting levels, as well as in HOMA-IR index after daily consumption of 300 mL of OJ during 60 days in a non-placebo-controlled clinical trial in healthy women [54]. 

However, to date, there are several clinical trials performed in different populations (healthy, obese, diabetic, or MetS) reporting no differences in glucose or insulin levels after chronic hesperidin or OJ consumption [55,56,57,58,59,60]. One study reported an increase in glucose levels in OJ-treated obese or overweight individuals, both in low and high hesperidin concentrations, which could be attributed to the daily addition to the diet of 500 mL dietary OJ during 12 weeks or to a decrease in insulin levels, which was also observed after the intervention [61]. 

### 2.2. Effects of Hesperidin on Lipid Profile and Adiposity 

The dysregulation of lipid and lipoprotein metabolism contributes to the pathogenesis of multitude of human diseases, including CVDs [62]. Several therapeutic strategies exist to modulate lipid metabolism and prevent the development of metabolic diseases, but these strategies present some inherent limitations. For instance, statin drugs, which have been widely used to improve lipid profile and reduce atherosclerotic risk, present well recognized side-effects such as myalgia, arthralgia, and temporary gastrointestinal upset [63]. Those patients presenting dyslipidemia associated with MetS are unable to reach their lipid treatment goals by the administration of statin drugs [64]. Considering this, flavonoids including hesperidin have emerged as new therapeutic agents that could prevent alterations regarding lipid metabolism. In this sense, hesperidin has been shown to be especially effective in modulating dyslipidemia associated with MetS, which is considered a major risk for atherosclerosis, by exerting lipid-lowering properties in animal models and humans [47,54,58,65,66,67,68,69]. Jung et al. investigated the effects of hesperidin on lipid regulation in C57BL/KsJ-*db*/*db* mice, a well-established model of obesity-induced T2D. The results of this study demonstrated that hesperidin (0.2 g hesperidin/kg diet) was effective in lowering the plasma free fatty acids (FFAs) and plasma and hepatic triglyceride levels after five weeks. Additionally, hesperidin reduced the hepatic fatty acid oxidation and carnitine palmitoyl transferase activity. Hesperidin effects on lipid regulation were attributable to a suppression of the hepatic fatty acid synthase, glucose-6-phosphate dehydrogenase, and phosphatidate phosphohydrolase activities and to an increase in the fecal triglycerides [43]. Furthermore, it was also demonstrated that hesperidin administration led to a decrease in plasma and hepatic cholesterol levels through a downregulation of the hepatic 3-hydroxy-3-methylglutaryl-coenzyme (HMG-CoA) reductase and acyl CoA: cholesterol acyltransferase (ACAT) activities [43]. Wu et al. demonstrated similar lipid-regulating effects with neohesperidin. Neohesperidin showed a potent hypolipidemic effect in HepG2 cells loaded with FFAs and reversed the pathological changes of lipid in the acute or chronic dyslipidemia mouse model. They suggested that neohesperidin regulates lipid metabolism in vivo and in vitro via fibroblast growth factor 21 (FGF21) and AMP-activated protein kinase/Sirtuin type1/Peroxisome proliferator-activated receptor gamma coactivator 1α signaling axis [51]. Hesperidin treatment has also been shown to reduce lipid accumulation in adipocytes derived from human mesenchymal stem cells by reducing lipogenesis and activating lipolysis [70]. Similar in vitro antiadipogenic effects have been observed in 3T3-L1 preadipocytes [71]. In addition, and related to lipid metabolism, Kim et al. have recently shown that hesperidin treatment increases *Uncoupling protein 3* (UCP3) expression in differentiated C2C12 myocytes, thus boosting energy consumption from lipids [72].

The beneficial effect of hesperidin on atherosclerosis development was demonstrated in a study conducted by Sun et al. using LDL receptor deficient (LDLr^−/−^) mice. The authors observed that hesperidin ameliorated high fat diet (HFD)-induced hyperlipidemia and suppressed HFD-induced hepatic steatosis, atherosclerotic plaque area, and macrophage foam cell formation. According to these results, Sun et al. suggested that hesperidin reduced atherosclerosis in part via amelioration of lipid profiles, inhibition of macrophage foam cell formation, its antioxidative effect, and anti-inflammatory action [47]. 

Therefore, results from in vitro and animal studies demonstrate a beneficial effect of hesperidin treatment on lipid profile, but these findings are in contrast with some human intervention studies. Thus, while the administration of glucosyl hesperidin to hypertriglyceridemic subjects for 24 weeks resulted in a clear reduction in plasma triglycerides and apolipoprotein B levels [73], in other studies, the administration of hesperidin capsules did not affect plasma total cholesterol, LDL-cholesterol, or triglyceride levels in moderately hypercholesterolemic individuals [74]. 

Adipose tissue plays an important role in storing lipid in the form of triglycerides, as well as secreting a variety of adipokines and cytokines [75]. However, adipose tissue dysfunction is a determinant cause for the development of obesity, an independent risk factor for CVDs [75,76]. In this sense, there are several studies demonstrating that hesperidin exerts beneficial effects on lipid accumulation and adiposity [71,72,77,78]. In animal models of obesity or MetS, a body-weight-reducing effect has been widely reported in response to hesperidin treatment [47,48,49,50,51], as well as a reduction in adipose tissue weight [25,48,50,51]. In contrast, Mosqueda-Solis et al. reported no significant changes in body weight after a daily hesperidin administration (100 mg/kg body weight) for eight weeks in Western-diet-fed rats, although hesperidin treatment resulted in a decreased size of adipocytes [78].

Similar to what has been observed in glucose and lipid metabolism, hesperidin or OJ treatment in obese or overweight individuals do not clearly reflect the effects observed in obesogenic animal models. Although Aptekmann and Cesar reported a significant reduction in body weight after daily consumption of OJ over 13 weeks in hypercholesterolemic subjects, no significant differences were observed between the intervention and control groups [68]. Rangel-Huerta and collaborators also observed a significant reduction in body weight after daily consumption of OJ over 12 weeks in obese or overweight subjects in a nonplacebo-controlled clinical trial [61]. By contrast, at least three studies reported no significant changes between control group and hesperidin or OJ groups in obese subjects [53,67,79].

### 2.3. Effects of Hesperidin on Blood Pressure and Endothelial Function

High blood pressure is one of the most significant risk factors for developing CVDs in all age groups [80]. In fact, it is known that a reduction of 10 and 5 mmHg in systolic blood pressure (SBP) and diastolic blood pressure (DBP), respectively, significantly decreases the relative risk of all major cardiovascular outcomes [81]. An extensive number of animal studies evaluating the cardioprotective role of hesperidin have shown its beneficial effects on high blood pressure [82,83,84,85,86]. The hypotensive effect after acute administration of hesperidin derivatives, hesperetin and glucosyl hesperidin (G-hesperidin), was demonstrated by Yamamoto et al. [24] in spontaneously hypertensive rats (SHR). In this study, a single oral dose of G-hesperidin (10 to 50 mg/kg) induced a dose-dependent reduction in SBP in SHR, but had no effect in control Wistar Kyoto rats (WKY), discarding possible hypotensive effects under normotensive conditions. The antihypertensive effect of hesperidin was suggested to be mediated by the vascular nitric oxide (NO) synthase pathway. Similar effects were reported by Liu et al., observing an increase in NO production in hesperetin-treated human endothelial cells [87]. In this sense, Ikemura et al. [83] reported a preventive effect of hesperidin and G-hesperidin against age-related increase in blood pressure. This preventive effect of hesperidin seemed to be mediated by an important increase in NO production in the groups supplemented with hesperidin or G-hesperidin and by an improvement in the endothelial function [83]. The long-term effects of hesperidin and G-hesperidin on blood pressure were also evaluated when administered to SHR and to WKY. Chronic oral administration for 25 weeks of hesperidin and G-hesperidin resulted in a decrease in blood pressure after 15 weeks of administration in SHR, while no changes occurred in WKY [82]. Recently, it was also demonstrated that chronic administration of hesperidin for eight weeks resulted in a significant reduction in SBP in cafeteria-fed rats, a well-stablished animal model for diet-induced hypertension [84,88]. They observed that chronic administration of hesperidin in these animals presenting diet-induced hypertension also resulted in lower secretion of inflammation and oxidative stress-related metabolites. A reduction in inflammation and oxidative stress could be the underlying mechanisms involved in hesperidin effects on blood pressure in these animals [84]. 

These findings suggest that a potential mechanism whereby hesperidin and its derivatives, including G-hesperidin and hesperetin, exert their beneficial effects on hypertension through their demonstrated antioxidant effect [20,83,89], enhancing NO bioavailability and protecting endothelial function from reactive oxygen species. Besides, several studies indicate that not only NO enhancement is involved in the antihypertensive effect exerted by hesperidin. The administration of hesperidin in SHR reduced blood pressure by reducing oxidative stress by the suppression of the renin–angiotensin system cascade [85]. In addition, hesperidin improved the reported oxidative stress observed under hypertensive conditions as a consequence of an overexpression of NADPH oxidase via suppression of this enzyme, which results in enhanced NO bioavailability [85,90,91].

Despite the beneficial effects observed in animal and in vitro studies, the results shown by human interventional studies are not consistent. Asgary et al. demonstrated that consumption of 500 mL/day of OJ decreased SBP and DBP in healthy subjects after four weeks [92]. Similar results were also reported by Morand et al. when evaluating the effect of daily consumption of OJ for four weeks in healthy volunteers [60]. In this study, it was stated that the beneficial effects on blood pressure maintenance induced by daily OJ consumption could be due to an improvement on endothelial function [60]. In another study, it was also demonstrated that six-week consumption of hesperidin improved blood pressure in T2D patients. The authors suggested that hesperidin exerts its beneficial effects via anti-inflammatory activity [93]. Furthermore, our group carried out a clinical study in which the beneficial effects of the consumption of OJ with natural hesperidin content and a hesperidin-enriched OJ on risk factors associated with CVDs, including its antihypertensive effects, in pre- and grade-1 hypertensive individuals were evaluated (submitted for its publication). However, the results of a crossover study that included individuals with MetS presenting prehypertension did not reveal changes in blood pressure after three-week supplementation with hesperidin [57]. Besides, a systematic review and meta-analysis of randomized controlled trials that evaluated the efficacy of hesperidin supplementation on blood pressure concluded that hesperidin intake is not associated with significant changes in blood pressure [94]. Similar results were reported by Plà et al. [95], concluding that hesperidin consumption effects on blood pressure were no conclusive. 

There are many hypotheses that could explain why hesperidin lacks a significant effect on blood pressure in humans, including its metabolization and absorption. In this sense, Yamamoto et al. [26] reported that the hesperidin metabolite hesperetin-7-O-glucuronide, but not hesperetin-3-O-glucuronide, was the responsible agent for the demonstrated antihypertensive effect. Therefore, not all hesperidin metabolites present the same biological effects when administered. In addition, few studies are available to clarify the pharmacokinetics of hesperidin [96]. In consequence, it might be possible that hesperidin does not reach the sufficient circulating concentrations that are needed for the regulation of blood pressure. 

In conclusion, the results from in vivo and in vitro studies point out that hesperidin represents a promising agent for the prevention and/or the treatment of CVDs. From these studies, it could be concluded that the potential mechanisms by which hesperidin exerts its beneficial effects include the regulation of gene expression and enzymatic activity of key proteins involved in pathways related to lipid and glucose metabolism, blood pressure control, and obesity development. Furthermore, it has been demonstrated the antioxidant and anti-inflammatory activities of hesperidin that may explain, at least in part, the observed beneficial effects on CVDs. However, despite all the evidences from in vitro and animal models, there is still controversy about whether hesperidin and their derivatives could contribute to ameliorating glucose homeostasis, lipid profile, adiposity, and blood pressure and thus reduce the cardiovascular risk, especially in humans (Figure 2). A possible explanation for the lack of conclusive results from human studies might be related to the presence of several important factors, including interindividual differences and external factors that impact the effectiveness of hesperidin in humans, including interindividual differences and external factors, with the variability of hesperidin bioavailability due to differences in intestinal microbiota composition and activity among individuals being a major factor. Although a low bioavailability of hesperidin has been described in animal studies [97,98], the variability in studies with experimental animals is much lower than that observed in clinical studies due to inbreeding, leading to a phenotypic uniformity between the animals. Furthermore, external factors such as diet, physical activity, or seasonality are much more controlled than in human studies, leading to lower variability in intestinal microbiota composition and activity. Further well-designed clinical trials to specifically examine the effects of hesperidin on CVD risk factors, considering the variability that exists in the response to treatment with hesperidin in humans, are necessary. In this sense, intestinal microbiota may play a role in this interindividual variability, as it has been shown to have a direct effect on the absorption and bioavailability of polyphenols, such as hesperidin.

## 3. Hesperidin and Intestinal Microbiota Interaction

The gastrointestinal tract is colonized by more than 10^11^ cells per mL of content, with the five main microbiota phyla being *Firmicutes*, *Bacteroidetes*, *Proteobacteria*, *Actinobacteria*, and *Verrucomicrobia* [99]. The intestinal microbiota is a complex ecosystem that varies between individuals and environmental conditions [100]. The gut microbiota plays an essential role in physical health status, and it is responsible for protecting the intestinal gut barrier mucosa against several pathogenic microorganisms, modulating the immune system, and producing some molecules that are beneficial for their hosts, such as vitamins or short chain fatty acids (SCFAs) [101]. 

### 3.1. The Gut Microbiota Assists in The Assimilation of Polyphenols

The gut microbiota present in the colon promotes absorption of some nutrients from the diet, including polyphenols, forming more bioactive and absorbable molecules than the original compounds directly consumed in food [102]. Both polyphenols and microbiota-derived metabolites may act on metabolic pathways and confer health benefits [103]. In addition, polyphenol metabolites derived from microbiota activity may contribute to the host with numerous health benefits [104], which are mainly summarized by two of their characteristics: (1) polyphenol antioxidant properties and (2) polyphenol antimicrobial capacity over various microorganisms, including some pathogenic bacterial species. 

### 3.2. Hesperidin: A Flavonol That May Promote a Healthier Profile of the Microbiota

Some clinical studies have demonstrated the role of polyphenols, supplemented in several food products, in maintaining the intestinal health and preserving microbial homeostasis by promoting the growth of beneficial bacteria and inhibiting the progression of pathogenic bacteria [105]. Indeed, polyphenols and hesperidin can modulate gut microbial composition or functionality, which affects the release of microbial-derived metabolites [106]. Flavonols are active inhibitors against some Gram-negative bacteria, such as *Prevotella* spp., *Porphyromonas gingivalis*, *Fusobacterium nucleatum*, *E. coli*, *Pseudomonas aeruginosa*, and *Clostridium spp.* [107,108] (Figure 3). In addition, hesperidin and other flavonols also inhibit the growth of some Gram-positive bacteria, such as *Staphylococcus aureus* and *Lactobacillus acidophilus* [107,108]. Besides their inhibitory capacity, phenolic compounds may modify gut microbiota by selectively promoting the growth of beneficial bacteria of the genera *Lactobacillus* or *Bifidobacterium* [109,110]. 

There is growing evidence suggesting that polyphenols may induce changes in the microbiota towards a more favorable composition and activity, including the production of SCFAs in the large intestine. These polyphenols derived metabolites have many known biological effects: (1) they are used as energy source for enterocytes [111]; (2) they improve gut barrier function [112]; and (3) they inhibit inflammation processes [112,113]. However, gut microbiota alteration, also known as dysbiosis, might also reduce the synthesis of SCFAs [106]. Concretely, butyric acid is used as an energy source for colonocytes and improves gut barrier integrity by promoting mucus secretion and increasing tight junction protein expression (essentially *zonulin* and *occludin*), which is translated to a reduced bacterial transport across the epithelium [114]. Propionic acid attenuates the secretion of several inflammatory cytokines and chemokines [115] and regulates key liver processes, such as gluconeogenesis [116]. Acetic acid may induce liver lipogenesis [116]. *Firmicutes* are the main butyrate producing bacteria in the human gut, especially *Clostridium leptum*, *Faecalibacterium prausnitzii*, *Roseburia spp*., and *Eubacterium rectale*. In addition, propionate and acetate are mostly produced by the *Bacteroidetes* phylum [117]. The negative effects of dysbiosis are partially compensated by hesperidin, as has been demonstrated in different animal studies, playing a dual role over both beneficial and harmful microbes [112]. Hesperidin selectively promotes the growth of some beneficial *Lactobacillus* species [113] and inhibits the growth of some harmful species, such as *Helicobacter ganmani* or *Helicobacter hepaticus* [112]. In contrast, hesperidin treatments also inhibit the growth of beneficial species as *Bifidobacterium pseudolongum* or *Mucispirillum schaedleri* [112] and promote the presence of harmful species as *Staphylococcus sciuri* and *Desulfovibrio* [112]. Additionally, hesperidin supplementation reduces gut inflammation by decreasing plasma levels of key proinflammatory cytokines (IL-1β, TNF-α, and IL-6) [112], reducing the colon mRNA expression of a key proinflammatory mediator, the inducible nitric oxide synthase (iNOS) [112], and increasing the small intestine IgA content [113]. Besides, hesperidin maintains intestinal gut barrier integrity, reducing key markers of intestinal integrity such as colon length, and plasma levels of intestinal fatty acid binding protein (iFABP) and lipid binding protein (LBP) [112]. In addition, hesperidin also promotes the expression of the three main tight junction components: *claudin 2*, *occludin*, and *zonula occludens-1* [112]. These results show the immunomodulatory actions of hesperidin on the gut and reinforce its role as a prebiotic; however, deeper studies of hesperidin effects on gut microbiota are necessary to completely understand these potential discrepancies.

### 3.3. The Gut Microbiota Dysbiosis is Associated with Increased CVD 

Gut microbiota dysbiosis and microbial infections are associated with several metabolic chronic disorders, including obesity, T2D, MetS, and CVDs. Fortunately, and as pointed in previous sections, polyphenols can promote a healthier state by improving the lipid and glucose metabolism [118], but at the same time, metabolic diseases may modify the gut microbiota composition [119]. In fact, the gut microbiota may regulate the development of metabolic disorders, not only by modulating nutrient absorption, but also by regulating intestinal barrier health, the low-chronic inflammation state and fat storage, processes that are tightly associated with the development of CVD risk factors [120,121]. In this line, numerous studies have demonstrated that obese subjects present a reduction in gut microbiota diversity compared to lean subjects [122] and, at the same time, subjects with low bacterial richness showed an increased dyslipidemia, adiposity, insulin resistance, and inflammatory state [120]. These evidences were confirmed when obese subjects were transplanted with the gut microbiota of lean donors [123]. After the transplantation, the gut microbiota of obese subjects presented an increase in bacterial diversity, with an associated increase in butyrate-producing bacteria and subsequent increase in insulin sensitivity [123]. Similar results were observed in animal models of transfected gut microbiota [120]. In this sense, germ-free mice transplanted with the microbiota of obese mice (*ob/ob*) donors, which harvest more energy than their lean counterparts, presented an increase in plasma leptin levels and elevated fasting glucose, which was translated with a systemic insulin resistance [124]. These results can be partly explained by the increased hepatic lipogenesis, which would be a consequence of the gut microbiota metabolization of indigestible polysaccharides into monosaccharides that posteriorly could be absorbed in the colon [120,125].

Moreover, obese subjects also exhibit an increase in *Firmicutes*, which leads to an increased *Firmicutes*/*Bacteroidetes* ratio [122]. A higher *Firmicutes*/*Bacteroidetes* ratio is associated with higher energy absorption from food, increased low-grade inflammation, and the development of obesity and insulin resistance [122,126]. In fact, gut microbiota dysbiosis is considered as a key contributor to the growing prevalence of obesity and associated cardiometabolic disorders, such as MetS or T2D [104]. Thus, *Akkermansia muciniphila*, a bacterial species increased by dietary polyphenols, was correlated with increased levels of some hormones such as glucagon-like peptide (GLP)1 and GLP2, which in turn promote insulin sensitivity [116]. In addition, *A. muciniphila* presence is reduced in obesity, and its levels are inversely related to adipose tissue mass and plasma glucose levels [125]. 

In gut dysbiosis, LPS, a key component of the Gram-negative bacterial membrane, promotes macrophage infiltration in adipose tissue, which in turn induces inflammation through the TLR4. LPS activates the inflammatory response by binding and activating TLR4, which triggers a signaling cascade that promotes the translocation of nuclear factor kappa-light-chain-enhancer of activated B cells (NF-kB) into the nucleus, where it stimulates the transcription of several inflammatory cytokines. However, some diet polyphenols, such as hesperidin, can increase the abundance of *Faecalibacterium prausnitzii*, which inhibits NF-kB activation and consequently attenuates the inflammatory response [9]. Increased LPS plasma levels disrupts the gut barrier permeability, probably due to reduced expression of key proteins that compose the tight junction, *zonulin* and *occludin*. These proteins contribute to form an impermeable intestinal epithelial barrier that prevents bacterial translocation and prevents harmful products derived from bacterial action reaching the bloodstream [115]. The intestinal mucosa may be considered as a complete immunological organ which contains immune cells, immunoglobulins (essentially IgA), and the microbiota [127]. 

In conclusion, studies investigating the effect of flavanones derived from oranges on the intestinal or fecal microbiota were mainly focused on their ability to inhibit the growth of pathogens, to increase beneficial species such as *Bifidobacterium spp.* and *Lactobacillus spp.*, and to stimulate the production of SCFAs (Figure 3). In fact, the relation between polyphenols and the gut microbiota is bidirectional. In the case of hesperidin, this flavanone can promote specific favorable bacterial species [128] and at the same time, hesperidin can be metabolized by specific microbiota bacteria [129,130].

### 3.4. Hesperidin Conversion to Hesperetin by the Microbiota Action

After food intake, hesperidin is poorly absorbed in the small bowel [10,29]. In fact, hesperidin absorption is highly dependent on the conversion to its active form, hesperetin, by the microbiota, and this phenomenon occurs mainly in the large intestine, where gut microbiota releases the rutinose moiety for further absorption by the colonocytes [129] (Figure 3). This conversion of hesperidin to hesperetin can be promoted by some specific microorganisms such as *Bifidobacterium pseudocatenulatum*, which can hydrolyze specific rutinose-conjugated polyphenols, which in turn release the aglycone form [130]. Therefore, *B. pseudocatenulatum* may potentially contribute to improving the bioavailability of hesperidin, which is particularly relevant because *Bifidobacteria* and polyphenols are significant components of the human diet, becoming a potential probiotic to improve the hesperidin absorption [29]. 

Deeper investigations are necessary to decipher the effects of hesperidin on the gut-associated lymphoid tissue, where hesperidin reaches first and, moreover, where it can interact with the gut microbiota, contributing to the crosstalk between gut bacteria and intestinal immune tissue. As stated above, polyphenols are extensively metabolized in the colon by the gut microbiota into several lower molecular weight and more absorbable parts, which might be responsible for the beneficial health effects [131], as occurs in the case of hesperidin. After absorption, these catabolites reach the bloodstream and are distributed systemically to the whole body, thereby altering the metabolome and influencing host health [84]. Summing up, thanks to the gut microbiota activity, flavanones, specifically hesperidin, are catabolized in the colon, increasing their bioavailability [132] and thus their potential beneficial effects on health. Among the species with the capability to produce these transformations are *Bifidobacteria* and *Lactobacillus* species, which are used as probiotics in many commercial food products and dietary supplements [114,133]. 

## 4. Hesperidin Bioavailability

For hesperidin to exert its beneficial effects, except for its effects on the colon, its metabolites and catabolites arising from the intestinal microbiota must be bioavailable and absorbed to be distributed through the bloodstream. In this sense, several bioavailability studies in humans showed a scarce and variable hesperidin absorption among individuals. Thus, in a clinical study with pure hesperidin intake of 89.1 mg, a cumulative urinary recovery in urine of 2% was observed [134]. Most studies on hesperidin bioavailability have been performed with OJ. In these studies, the consumption of between 250 mL and 1250 mL OJ resulted in a total flavonoids metabolites recovery (including hesperidin and naringin metabolites) in urine between 2.9% and 24% [30,31,135,136,137,138] (Table 1).

Hesperidin bioavailability studies show that there is a high interindividual variability in its bioavailability [30,31,134,135,138,139]. In fact, the stratification of individuals as high, intermediate, or low hesperidin metabolite excretors has been proposed, as assessed by urinary excretion of hesperidin metabolites [30,31,134,135,138,139]. 

The clinical studies discussed above described hesperidin bioavailability by quantifying hesperidin metabolites excreted in urine in the form of glucuronide or sulphated conjugates relative to the amount of hesperidin consumed. Nevertheless, a substantial portion of hesperetin is further metabolized by the microbiota present in the colon to bioavailable catabolites, including highly specific catabolites of hesperetin such as 3-(3′-hydroxy-4′-methoxyphenyl) propionic acid (HMPPA) as well as less specific catabolites like hippuric acid, 4-hydroxyhippuric acid and 3-(3′-hydroxyphenyl) hydracrylic acid (HPHPA), which may also result from other phenolic sources [31]. Interestingly, it has been described that bioavailability of hesperidin increases considerably if the catabolites generated by the intestinal microbiota, such as HMPPA, are considered. In fact, different bioavailability studies in humans that included the measure of these catabolites observed a wide spectrum of urinary excretion levels of hesperidin metabolites and catabolites relative to the consumed flavonoid of 45.9% [31], 64.2% [30], and 100% [137] (Table 1). As a result, quantifying metabolites and catabolites generated by the microbiota provides a new perspective in the bioavailability of hesperidin.

### 4.1. Factors Affecting Hesperidin Bioavailability

#### 4.1.1. Microbiota Composition and α-L-Rhamnosidase Activity

The low bioavailability of hesperidin has been attributed to the sugar moiety (the rutinose disaccharide) conjugated to the hesperetin molecule, which hampers absorption in the small intestine [27,143], leading to the vast majority of hesperidin ingested having to be metabolized in the colon by the α-rhamnosidase and the β-glucosidase activity of intestinal microbiota to form hesperetin, its aglycone form [144]. It has been described that α-rhamnosidase is the limiting step of the formation of hesperetin due to low levels of this enzymatic activity in the intestinal microbiota [130]. Thus, it is conceivable that intestinal microflora composition may have a large impact on hesperidin bioavailability due to variations in the activity of the α-rhamnosidase enzymes. Interestingly, a high negative relationship has been observed between excretion levels of hesperidin metabolites and excretion levels of hesperetin catabolites from the intestinal microbiota, such as HMPPA [31], indicating that the microbiota of high and low hesperidin excretors have different activities associated with hesperidin metabolism. However, despite the potential effects of α-rhamnosidase activity on hesperidin bioavailability and its biological effects, there are no available studies to clearly demonstrate these effects. Therefore, further in vivo studies in humans are needed to clarify the effect of α-rhamnosidase activity on the bioavailability of hesperidin.

#### 4.1.2. Stereochemical Properties of Hesperidin

The hesperidin molecule has a chiral carbon that generates two diastereoisomers, –R and –S; however, in nature, the predominant form is the –S diastereoisomer [145]. Hesperidin is present in the fresh fruit products, including OJ, in a S:R ratio of at least 92:8 in favor of the 2S-epimer [145]. 

The stereochemical properties of flavonoids have been reported to influence their bioavailability [146]. Specifically for hesperidin, the effects of its stereochemical properties on plasma and urinary kinetics of hesperetin have been described [147,148], and may thus affect both the intestinal metabolism and transport of hesperetin as well as its biological effects [149]. In an in vivo study, the administration of racemic hesperetin to rats demonstrated that R-hesperetin had a significant 3.3-fold higher area under the serum concentration–time curve (AUC), a 1.9-fold longer half-life, and a 2.3-fold higher cumulative urinary excretion compared to S-hesperetin [148]. Furthermore, Yez et al. observed that oral administration of racemic hesperidin to a single rat revealed a slightly (~15%) increased cumulative 24 h urinary excretion of R-hesperetin compared to S-hesperetin [147]. In an in vitro study with human small intestinal microsomes, the authors observed a higher affinity and capacity towards S-hesperetin resulting in an overall 5.2-fold higher catalytic efficiency for the formation of S-hesperetin glucuronides as compared to R-hesperetin glucuronides [149]. This is important because although hesperidin naturally exists mainly as the 2S-epimer, which upon intake is subsequently transformed into S-hesperetin, practically all research on hesperidin and hesperetin using “pure” compounds is on racemic mixtures, because the vast majority of current hesperidin products are commercially available as a mixture of both diastereoisomers [149]. Due to these scenarios, there are companies that are investigating producing hesperidin products consisting of 100% S-hesperidin.

#### 4.1.3. Food Matrix and Food Processing

The bioavailability of hesperidin can be influenced by the food matrix in which it is consumed. In this sense, the term bioaccessibility refers to the fraction of a compound that is released from its matrix in the gastrointestinal tract and thus becomes available for intestinal absorption. In a clinical study by Mullen et al., the impact of a full-fat yogurt on the bioavailability of OJ flavanones was investigated. Hesperidin levels in urine and plasma were measured after the consumption of 250 mL of OJ, with and without 150 mL of full-fat yogurt. The results demonstrated that although the quantity of flavanone metabolites excreted 0–5 h after OJ ingestion was significantly reduced by yogurt, over the full 0–24 h urine collection period, the amounts excreted were not affected by the addition of yogurt to the drink. The authors concluded that the full-fat yogurt had little effect on the bioavailability of the OJ flavanones probably due to the low amount of fats in the yogurt [140]. 

Related to the food matrix factor, the solubility of a given metabolite is a requirement to enter the systemic circulation and exert a physiological effect. For hesperidin, a low solubility has been described, especially in aqueous systems [150]. In a crossover clinical study with 10 volunteers, Vallejo et al. evaluated the effect of hesperidin concentration and solubility of orange beverages on its bioavailability. Participants consumed five different beverages with different hesperidin concentrations. The results showed that hesperidin excretion and maximal concentration (Cmax) in plasma were correlated with the soluble hesperidin concentration in juice, whereas no correlation was observed with the total hesperidin intake. The authors concluded that the solubility of hesperidin in the juice was a key factor for the bioavailability [139].

In a study that evaluated the in vitro bioaccessibility of hesperidin in different matrix sources, it was observed that the bioaccessibility of hesperidin increased significantly upon juice extraction compared to orange segments [151]. The authors concluded that the lower flavonoid levels in OJ as compared to orange segments might be less relevant regarding their intestinal absorption, because low flavonoid solubility may be the limiting factor [151]. In a clinical study evaluating the bioavailability of hesperidin from orange fruit and from OJ, despite the higher hesperidin dose delivered with the orange fruit, urinary hesperetin excretion did not differ from that observed after the consumption of OJ, suggesting that release, absorption, and metabolism of dietary flavanones are saturated when intake exceeds a certain limit. Another possible explanation was the entrapment of hesperidin within the fiber-rich matrix of orange fruit [31]. Supported by these findings, it is assumed that the higher hesperidin level in orange fruits compared to OJ offers only a limited nutritional benefit. Nevertheless, in another clinical study by Brett et al., no differences were observed in bioavailability, based on total urinary hesperetin excretion of human subjects after consumption of orange fruit and OJ matrices [138].

Hesperidin in OJ exists as both soluble in the juice serum and precipitated in the juice cloud. It has been suggested that hesperidin associated with the juice cloud may be available to enzymatic action in the gastrointestinal tract at different rates than the soluble form [152]. Furthermore, it has been demonstrated that the distribution of hesperidin in OJ is influenced by commercial juice processing and storage techniques [141,152,153,154], with total concentrations of soluble hesperidin being higher in hand squeezed OJ than in commercially processed OJ [141,153,154], while freezing and cold storage of processed juice decreases hesperidin solubility [152]. However, a clinical study that measured the bioavailability of hesperidin after single doses of hand-squeezed OJ or commercially processed OJ in healthy humans showed no statistically significant difference in the percentage of urinary-excreted hesperidin between both different styles of OJ products [142], suggesting that absorption of the OJ flavanones was not appreciably influenced by the distributions of soluble and precipitated forms [142].

Efforts have been made to overcome the drawbacks of poorly water-soluble hesperidin to enhance its absorption. In this sense, the use of nanotechnology to encapsulate hesperidin represents a promising strategy to circumvent hesperidin physicochemical and bioavailability constraints, since it might enhance hesperidin’s solubility and absorption [155]. In fact, in a clinical study that evaluated the effect of hesperidin encapsulation and particle size reduction on hesperidin bioavailability, it was observed that both hesperidin micronization and encapsulation increased hesperidin bioavailability compared to conventional hesperidin. The authors concluded that particle size reduction and hesperidin dispersion are two ways to enhance its bioavailability. Furthermore, the results suggested that micronization can be used to overcome the need for gut microbiota α-rhamnosidase hydrolysis by enhancing hesperidin solubility and reducing particle size to facilitate the interaction with intestinal cells and gut microbiota [134].

In recent years, several delivery nanocarrier-based formulations have been developed to modulate the release of bioactive compounds, including polyphenols such as catechins, quercetin, eugenol, epigallocatechin, curcumin, and tea polyphenols, demonstrating improvements in the solubility of these bioactive molecules, improving their bioavailability, absorption, and biological effects [155,156]. In this sense, in the past years, nanosuspensions, polymeric nanoparticles, nanocapsules, nanofibrous scaffolds, nanoemulsions, and nanoliposomes, among others, have been investigated to deliver polyphenols and improve its bioavailability and bioactivity [155,157]. However, concerns related to the nanoencapsulation of polyphenols have been reported, due to their varying structures, solubility, and fast oxidation under basic conditions [155]. Furthermore, various wall materials, preparation methods, encapsulation processes, and release mechanisms, as well as several main factors including pH values, temperatures, particle sizes, and additives, can strongly influence the encapsulation processes and efficacy [158]. Therefore, efforts should be made to evaluate the effects of nanoencapsulation on the metabolism, bioavailability, and efficacy of hesperidin. 

## 5. Future Remarks

Large interindividual variability regarding hesperidin bioavailability may explain some of the discrepancies observed among the results from different clinical trials reporting the effects of hesperidin on CVD risk factors. Volunteer stratification into high, medium, and low urinary metabolites excretors or metabotypes may explain, at least partly, this large interindividual variability. In turn, this interindividual variability can be determined by several factors, such as age, sex, genetics, or gut microbiota. Personalized nutrition is a way to address interindividual differences, since it aims to deliver nutritional intervention or nutritional advice suited to the particular characteristics of each person, in order to maximize the beneficial effects of diet and dietary compounds on health [159].

Two approaches to reach personalized nutrition have been defined. Thus, personalized nutrition, or also named individually tailored nutrition, attempts to deliver nutritional intervention or nutritional advice suited to each individual, whereas stratified nutrition, or tailored nutrition, attempts to group individuals with shared characteristics and to deliver nutritional intervention or nutritional advice that is suited to each group. Tailored nutrition has been based primarily on the analysis of genetic variations to cluster individuals according to genetic set up. However, the benefits for public health have been limited [160]. In this context, the use of metabolomics and metabolite profile analysis allows one to tackle personalized nutrition thoroughly, since the interaction and outcome of different factors, such as genes, diet, microbiota, and environmental factors, are included in metabolite profiles, provide complete information on the biological processes of the organism [161]. The concept of metabolic phenotype, or metabotype, refers to the combination of specific metabolites to classify individuals into groups or clusters based on a similar metabolic phenotype. In the context of nutrition, metabolic phenotyping allows one to examine the response of individuals to dietary interventions and to deliver dietary advice adapted to individuals depending on their specific metabotype [161].

As an example of the use of metabotyping for individual clustering to personalize nutrition, in a clinical study, the authors observed that individuals may be classified based on the metabotype associated with the ellagic acid polyphenol (metabotype A, B, or 0) according to the metabolite profile excreted in their urine after ingestion of this polyphenol [162,163]. By this classification, supplementation of the diet with a pomegranate extract, which is rich in ellagic acid, was able to lower blood cholesterol levels in those individuals with a specific profile of metabolites of the metabotype B, while no significant differences on the cholesterol concentration were observed when the effects were analyzed in all the study subjects. In addition, the distribution of these metabotypes among the population was different depending on their state of health, with a higher frequency of metabotype B in obese people and those with MetS [162].

Given that the intestinal microbiota plays an important role in hesperidin metabolism and absorption through α-rhamnosidase activity, and that the activity of this enzyme can vary considerably depending on the composition of the microbiota, the metabotype associated with hesperidin consumption will be closely linked to the microbial profile. However, the relation between the hesperidin metabotype (low, medium, or high urine metabolites excretors) and α-rhamnosidase activity has not been studied in humans. In addition, these different metabotypes have not been associated with specific enterotypes (e.g., a profile of the characteristic microbiota), which would help to explain the role of the microbiota in the bioavailability of hesperidin. 

In conclusion, animal and human studies are necessary to clarify the relationship between the composition of the intestinal microbiota, the activity of α-rhamnosidase, the metabotypes of hesperidin consumption, and the effects of this flavonoid on human health. Furthermore, clinical trials to evaluate the beneficial effects of hesperidin consumption on health should be considered, as well as the classification of individuals according to metabotypes.

## Figures and Tables

**Figure 1 nutrients-12-01488-f001:**
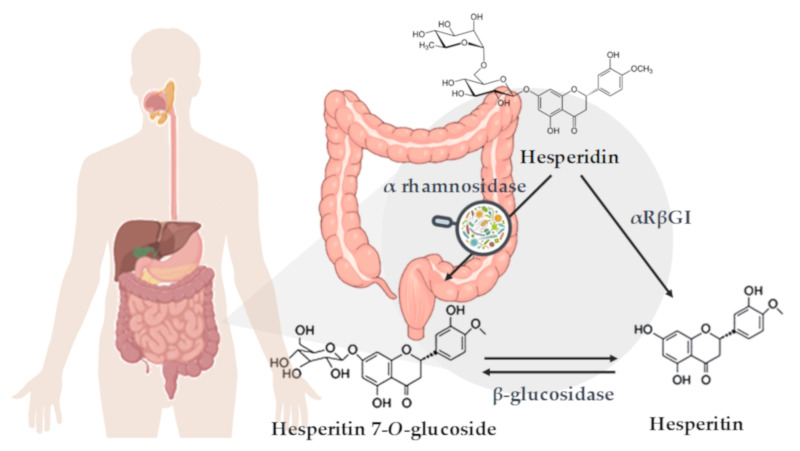
Schematic representation of hesperidin metabolization in the colon. Enzymatic deglycosylation of hesperidin to yield hesperetin: via hesperetin-7-O-glucoside by two specific monoglycosidases, α-rhamnosidase and β-glucosidase, and via one-step deglycosylation through α-rhamnosyl-β-glucosidase (αRβGl).

**Figure 2 nutrients-12-01488-f002:**
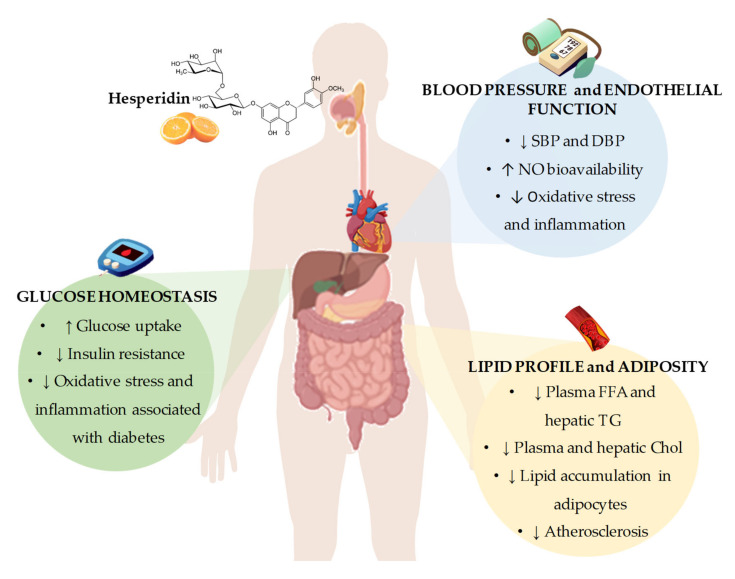
Summary of the most representative effects of hesperidin consumption and its derivatives on cardiovascular risk factors, including glucose homeostasis, blood pressure and endothelial function, and lipid profile and adiposity. SBP: systolic blood pressure; DBP: diastolic blood pressure; NO: nitric oxide; FFA: free fatty acids; TG: triglycerides; Chol: cholesterol.

**Figure 3 nutrients-12-01488-f003:**
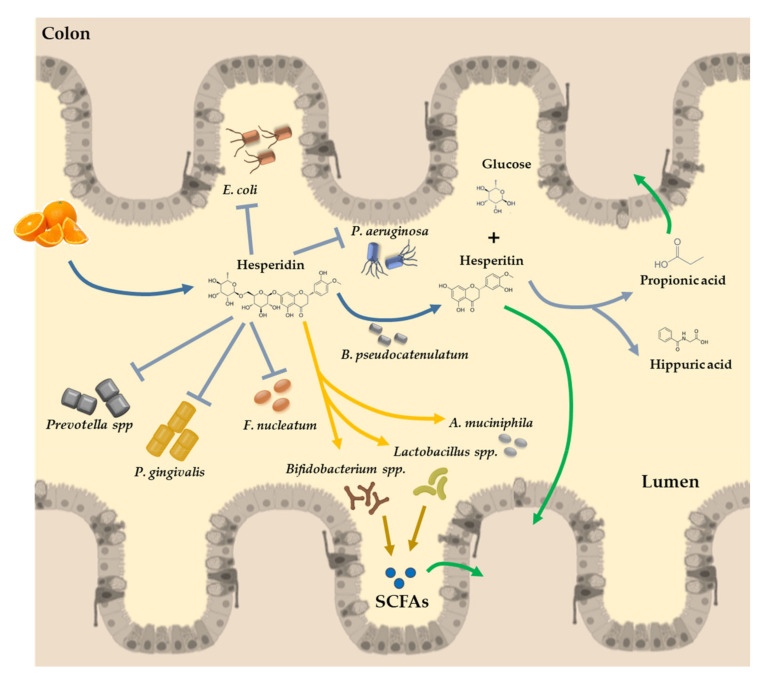
Illustrative diagram of hesperidin absorption in the colon. The flavones present in oranges reach the colon almost unchanged in their structure. In the lumen of the colon, hesperidin is converted to its active form by the α-rhamnosidase activity of the microbiota (*Bifidobacterium pseudocatenulatum*), releasing the rutinose moiety and hesperetin for further absorption by the colonocytes. In the colon, hesperidin promotes the growth of some beneficial bacteria species, with a key role in the SCFA production (*Bifidobacterium* spp., *Lactobacillus* spp., or *Akkermansia muciniphila*). SCFAs are absorbed with healthy effects in the permeability of the gut barrier and in distal organs and tissues. Moreover, hesperidin has other beneficial effects by inhibiting the proliferation of detrimental bacteria, such as *Escherichia coli*, *Pseudomonas aeruginosa*, *Prevotella* spp., *Porphyromonas gingivalis*, and *Fusobacterium nucleatum*, among others. SCFAs: short chain fatty acids.

**Table 1 nutrients-12-01488-t001:** Hesperidin bioavailability expressed as percent urinary excretion in human intervention studies.

Administration Form	Hesperidin Dose (µmol)	Population	Measured Metabolites	Relative Urinary Excretion (%)	Reference
Conventional Hesperidin Micronized Hesperidin Encapsulated Hesperidin	146.1 146.1 146.1	18 healthy subjects (10 men and 8 women)	Hesperetin glucuronides and sulfates metabolites	2.2 ± 0.3 4.3 ± 0.9 5.4 ± 0.8	[134]
400–760 mL commercial OJ	206.6 ± 42.6	8 healthy subjects (3 men and 5 women)	Total hesperetin metabolites	5.3 ± 3.1	[135]
500 mL commercial OJ	180.3 ± 6.1	5 healthy men	Total hesperetin metabolites	4.3 ± 1.2	[136]
1000 mL commercial OJ	360.6 ± 12.1			6.4 ± 1.3	
500 mL commercial OJ	250	10 males in trained conditions	Hesperetin glucuronides and sulfates metabolites/catabolites	3.8 ± 2.3/51	[30]
		10 males in detrained conditions		4.8 ± 2.8/59	
250 mL pulp enriched OJ	348	12 healthy subjects (6 men and 6 women)	Hesperetin glucuronides and sulfates metabolites/catabolites	17.5 ± 2.0/26.2	[137]
400 g orange fruit	1477 ± 88	11 healthy subjects	Total hesperetin metabolites/catabolites	1.5 ± 0.5/20.3	[31]
719 g commercial OJ	636 ± 17			2.9 ± 1.1/40.4	
150 g orange fruit	130.6 ± 29.0	20 healthy subjects (10 men and 10 women)	Total hesperetin metabolites	4.3 ± 3.4	[138]
300 g commercial orange fruit	117.7 ± 13.3			4.6 ± 3.1	
400 mL commercial OJ	191.5 ± 2.0	10 healthy subjects	Hesperetin glucuronides	5.4 ± 1.2	[139]
400 mL commercial OJ	352.8 ± 5.2	(5 men and 5 women)		1.7 ± 0.4	
400 mL pulp enriched OJ	461.0 ± 2.0	10 healthy subjects	Hesperetin glucuronides	1.0 ± 0.5	[139]
400 mL OJ with flavanone extract	722.6 ± 10.5	(5 men and 5 women)		4.6 ± 1.0	
400 mL water with flavanone extract	339.7 ± 2.0			8.9 ± 2.9	
Commercial OJ Commercial OJ supplemented with hesperidin	100.0 314.7	16 healthy subjects (8 men and 8 women)	Total hesperetin metabolites	4.6 ± 1.8 8.9 ± 3.8	[27]
250 mL commercial OJ 250 mL commercial OJ and 150 mL full fat yogurt	168 168	8 healthy subjects (4 men and 4 women)	Hesperetin glucuronides and sulfates	6.3 ± 2.0 6.4 ± 2.0	[140]
400 mL hand squeezed OJ	62.0	18 healthy subjects	Hesperetin glucuronides and	8.1 ± 1.4	[141]
400 mL high pressure homogenized OJ	169.6	(10 men and 8 women)	sulfates	4.8 ± 1.1	
400 mL pasteurized OJ	184.9			3.3 ± 0.5	
786 mL processed OJ	199.2 ± 60.8	24 healthy subjects (12 men and 12 women)	Hesperetin glucuronides and sulfates	4.1 ± 3.3	[142]
786 mL fresh OJ	60.8 ± 5.2			3.8 ± 2.2	

OJ, orange juice.

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
