# Peer review of "Effect of Hesperidin on Cardiovascular Disease Risk Factors: The Role of Intestinal Microbiota on Hesperidin Bioavailability"

_nutrients, 2020, doi:10.3390/nu12051488_

Round 1
Reviewer 1 Report
Hesperidin effects on cardiovascular disease risk 3 factors: the role of intestinal microbiota on hesperidin 4 bioavailability
Review:
- While the authors describe an interesting relationship between hesperidin and several cardiovascular outcomes and have tried to attribute the differential benefits in humans as a result of differences in bioavailability as well as influence of gut microbiota, overall this seems to be a rather descriptive manuscript.
- Mechanisms for the cardiovascular and other metabolic risks have multiple etiologies and no global evidence-based hypothesis is rationalized, tempering enthusiasm for this work.
- The authors do note that some animal studies have reported beneficial effects, but these have not translated into clinical outcomes. While it is well know that several molecules or therapeutics may have success in animals but fail in humans, the reviewer is not convinced why similar bioavailability and microbial clonal shifts are not occurring in animals as these are relatively preserved mechanisms across species. A greater description of these in animals would be recommended.
- The authors also note that polyphenols may induce changes in the microbiota towards a more favorable composition including the production of SCFAs in the large intestine. Hesperidin can modulate bacterial species, provide energy source for enterocytes, improve gut barrier function, and inhibit inflammatory processes. To truly attribute these effects to hesperidin one would like to see additional studies where blocking of these individual pathways results in a lack of response to Hesperidin. The reviewer would suggest that those are further detailed in this manuscript.
- The idea of personalized medicine is not novel and has been described previously. It is quite possible that the variable responses with hesperidin are secondary to individual differences but the same can be said for almost any therapeutic or even a diagnostic test, thus not specific to hesperidin.
- Conclusions as noted by the authors are not supported in an objective manner and it would have helped to use a standardized mechanism for data reporting for systematic reviews like PRISMA.
- Several syntax and grammatical errors which need to be corrected.
Author Response
RESPONSE TO REVIEWER 1
Thank you so much for your kind and professional suggestions. We really appreciate all your comments to improve our manuscript.
You can find all your suggestions with our comments, reviewed point by point, below.
- While the authors describe an interesting relationship between hesperidin and several cardiovascular outcomes and have tried to attribute the differential benefits in humans as a result of differences in bioavailability as well as influence of gut microbiota, overall this seems to be a rather descriptive manuscript.
Answer (A): Thank you for your comment and as the reviewer points out, our objective was to carry out a descriptive review of the involvement of Hesperidin in cardiovascular diseases.
- Mechanisms for the cardiovascular and other metabolic risks have multiple etiologies and no global evidence-based hypothesis is rationalized, tempering enthusiasm for this work.
A: We agree with the reviewer’s comment and we have added this information in the “2.3. Effects of hesperidin on blood pressure and endothelial function” section, lines 277-283 of the new manuscript version:
“the results from in vivo and in vitro studies point out that hesperidin represents a promising agent for the prevention and/or the treatment of CVDs. From these studies it could be concluded that the potential mechanisms by which hesperidin exert its beneficial effects include the regulation of gene expression or also enzymatic activity of key proteins involved in pathways related to lipid and glucose metabolism, blood pressure control and obesity development. Furthermore, it has been demonstrated the antioxidant and anti-inflammatory activities of hesperidin that may explain, at least in part, the observed beneficial effects on CVDs.”
- The authors do note that some animal studies have reported beneficial effects, but these have not translated into clinical outcomes.
While it is well know that several molecules or therapeutics may have success in animals but fail in humans, the reviewer is not convinced why similar bioavailability and microbial clonal shifts are not occurring in animals as these are relatively preserved mechanisms across species.
A greater description of these in animals would be recommended.
A: We agree with the reviewer that a more detailed description to explain the potential reason between the discrepancies observed between animal and human studies on hesperidin bioavailability and effectiveness is necessary.
For this reason, we have added this information at 286-295 lines of the new manuscript version:
“A possible explanation for the lack of conclusive results from human studies might be related to the presence of several important factors, including interindividual differences and external factors, that impact the effectiveness of hesperidin in humans, being the variability of hesperidin bioavailability due to differences in intestinal microbiota composition and activity among individuals a major factor.”
- The authors also note that polyphenols may induce changes in the microbiota towards a more favorable composition including the production of SCFAs in the large intestine. Hesperidin can modulate bacterial species, provide energy source for enterocytes, improve gut barrier function, and inhibit inflammatory processes. To truly attribute these effects to hesperidin one would like to see additional studies where blocking of these individual pathways results in a lack of response to Hesperidin. The reviewer would suggest that those are further detailed in this manuscript.
A: We appreciate and agree with the reviewer comment that a more detailed description about the effects of Hesperidin in the modulation of bacterial composition and in the gut metabolism (inflammation, gut barrier integrity and source of energy for colonocytes) are necessary.
For that, in the new version, it has been included some animal studies that focus on the individual pathways that regulate these mechanisms (Estruel-Amades et al. 2019; den Besten et al. 2013; and Liu et al. 2020). Concretely, it has been included the following paragraph in the “3. Hesperidin and intestinal microbiota interaction” section; lines 347-363:
“The negative effects of dysbiosis are partially compensated by hesperidin as has been demonstrated in different animal studies, playing a dual role over both beneficial and harmful microbes [112]. Hesperidin selectively promotes the growth of some beneficial Lactobacillus species [113]; and inhibits the growth of some harmful species as Helicobacter ganmani or Helicobacter hepaticus [112]. In contrast, hesperidin treatments also inhibit the growth of beneficial species as Bifidobacterium pseudolongum or Mucispirillum schaedleri [112] and promotes the presence of harmful species as Staphylococcus sciuri and Desulfovibrio [112]. Additionally, hesperidin supplementation reduces gut inflammation by decreasing plasma levels of key pro-inflammatory cytokines (IL-1β, TNF-α and IL-6) [112], reducing the colon mRNA expression of a key pro-inflammatory mediator, the inducible nitric oxide synthase (iNOS) [112], and increasing the small intestine IgA content [113]. Besides, hesperidin maintains intestinal gut barrier integrity, reducing key markers of intestinal integrity such as colon length, and plasma levels of intestinal fatty acid binding protein (iFABP) and lipid binding protein (LBP) [112]. In addition, hesperidin also promotes the expression of the three main tight junction components: claudin 2, occludin and zonula occludens-1 [112]. These results show the immunomodulatory actions of hesperidin on the gut and reinforce its role as a prebiotic, however, deeper studies of hesperidin effects on gut microbiota are necessary to completely understand these potential discrepancies.”.
- The idea of personalized medicine is not novel and has been described previously. It is quite possible that the variable responses with hesperidin are secondary to individual differences but the same can be said for almost any therapeutic or even a diagnostic test, thus not specific to hesperidin.
A: We agree with the reviewer’s opinion that personalized nutrition is not specific to hesperidin treatment. To describe this message in a clearer and more general way, we have changed the phrase on line 591-594 of new manuscript version:
“Personalized nutrition is a way to address these inter-individual differences, since it aims to deliver nutritional intervention or nutritional advice suited to the particular characteristics of each person, in order to maximize the beneficial effects of diet and dietary compounds on health”
to:
“Personalized nutrition is a way to address inter-individual differences, since it aims to deliver nutritional intervention or nutritional advice suited to the particular characteristics of each person, in order to maximize the beneficial effects of diet and dietary compounds on health”
- Conclusions as noted by the authors are not supported in an objective manner and it would have helped to use a standardized mechanism for data reporting for systematic reviews like PRISMA.
A: This manuscript is a literature review, not a systematic or meta-analysis review, and therefore this manuscript does not follow PRISMA guidelines.
In fact, several literature reviews are published in the Nutrients journal, following a similar structure. Just to have as example:
Nutrients 2020, 12(5), 1265; https://doi.org/10.3390/nu12051265
Nutrients 2020, 12(5), 1248; https://doi.org/10.3390/nu12051248
- Several syntax and grammatical errors which need to be corrected.
A: We thank to the reviewer for this comment and we apologize for these mistakes. Following the reviewer’s recommendation, a native English speaker has performed an extended revision of the manuscript.

Reviewer 2 Report
This is a comprehensive review on a topic of widespread scientific interest and immense available literature, specifically the potential associations between nutraceutics (MeSH term ‘Dietary Supplements’, n=74431), intestinal microbiota (MeSH term ‘Microbiota’, n=31912), risk factors (MeSH term ‘Risk factors’, n= 810206), and cardiovascular disease (MeSH.term ‘Cardiovascular diseases’, n=2354503). This review focuses on hesperidin, a flavonone mainly present in citrus fruits, its potential therapeutic use in the field of cardiovascular diseases, and the role of intestinal microbiota as potential mediator of the important interindividual differences in hesperidin bioavailability.. The article is comprehensive, well structured, well written, the reference section is accurate, and the illustrations are of high quality. The reader will appreciate the possibility to be introduced in a really complex and debated world, but in a clear and friendly manner. I am sure this article will be very appreciated by a broad audience of professionals.
Author Response
Reviewer 2
Comments and Suggestions for Authors
This is a comprehensive review on a topic of widespread scientific interest and immense available literature, specifically the potential associations between nutraceutics (MeSH term ‘Dietary Supplements’, n=74431), intestinal microbiota (MeSH term ‘Microbiota’, n=31912), risk factors (MeSH term ‘Risk factors’, n= 810206), and cardiovascular disease (MeSH.term ‘Cardiovascular diseases’, n=2354503). This review focuses on hesperidin, a flavonone mainly present in citrus fruits, its potential therapeutic use in the field of cardiovascular diseases, and the role of intestinal microbiota as potential mediator of the important interindividual differences in hesperidin bioavailability.. The article is comprehensive, well structured, well written, the reference section is accurate, and the illustrations are of high quality. The reader will appreciate the possibility to be introduced in a really complex and debated world, but in a clear and friendly manner. I am sure this article will be very appreciated by a broad audience of professionals.
Thank you so much for your kindly constructive and professional suggestions. We really appreciate all your comments about our manuscript.

Reviewer 3 Report
Thank you for the opportunity to review this article. This is a review that aims to describe the effects of hesperidin consumption on CVD prevention, and the potential role of the gut microbiota. Although it is obvious that the authors put a great amount of information together, the manuscript miss a good structure: the abstract seems an extension of the introduction; the introduction is quite long and it focus too much on information on the structure of the hesperidin, while lacking to provide a good rationale for the study. There is no methods section (?). I may be missing something, but the authors do not seem to follow the instructions defined by Nutrients on how to prepare a review.
Abstract
The introduction of the abstract is rather long, and the authors present no results in the abstract and seem to finalize with their hypotheses. Please follow the basic structure of a study in your abstract: “introduction, methods, results, conclusion”. Even if these words are not explicit written in the abstract, the information should be there.
Introduction
The introduction is rather long. The authors could try to cut non-essential information to keep it shorter.
Line 34: “currently in the world”?
I suggest the authors to have a look on the references below, it may be useful for their introduction:
Collaborators GBDD. Health effects of dietary risks in 195 countries, 1990-2017: a systematic analysis for the Global Burden of Disease Study 2017. Lancet. 2019. doi:10.1016/S0140-6736(19)30041-8.
WHO - World Health Organization, editor. Vienna Declaration on Nutrition and Noncommunicable Diseases in the Context of Health 2020. WHO Ministerial Conference on Nutrition and Noncommunicable Diseases in the Context of Health 2020; 2013 4-5 July Vienna, Austria.
Line 40: in which context is diet considered an environmental factor rather than an individual factor?
Lines 42 - 45: Please be more specific about what you consider treatment and prevention. For instance, by ‘common therapies for patients with CDV disorders’, do you mean treatment for hypertension based on drugs? And these ‘alternative treatments’, are they prevention strategies for CDV? Treatment of CDV? Or hypertension? The history line is not easy to follow in those sentences, perhaps use example to better explain.
Line 61: harmless when considering the upper-bound limits of indicated maximum consumption, according to the Dietary Reference Intakes (DRI)?
Lines 63 – 68: I would suggest the authors to synthesise these explanations. The information here can be cut down to three lines.
Lines 69 – 74: Since you present figure 1, the information described here on the chemical structure of the hesperidin can be deleted.
Lines 80 – 95: this paragraph also brings too much information, please stick to the essential information that will help the reader to understand the motivation for your study.
In the abstract, which now seems like an extension of their introduction, the authors say that hesperidin has been shown to potentially contribute to obesity and CVD prevention in models, but inconsistent or null information on a protective effect of hesperidin was available for humans. However, to my surprise, there is no information about that in the introduction, which to my understanding is the very rationale for conducting a review. I urge the authors to substantially reduce the information on the chemical mechanisms and structure of hesperidin and improve their rationale on why this review is needed. What is the findings from studies in animal models? What kind of information is available from human studies? Why is it important to review the literature on this subject?
Methods?
I do not understand the structure of this manuscript. I am reviewing this manuscript assuming this is a systematic review. I checked the Nutrients’ instructions for authors (https://www.mdpi.com/journal/nutrients/instructions#submission), and noticed they recommend review articles to follow the PRISMA guidelines http://prisma-statement.org/documents/PRISMA%202009%20checklist.pdf, which this manuscript does not follow. There is for instance no information about the database that the authors searched, what were the terms of the search, there is not even a Methods section.
Author Response
Reviewer 3
Comments and Suggestions for Authors
Thank you for the opportunity to review this article. This is a review that aims to describe the effects of hesperidin consumption on CVD prevention, and the potential role of the gut microbiota.
Thank you so much for your kind and professional suggestions. We really appreciate all your comments to improve our manuscript.
You can find all your suggestions with our comments, reviewed point by point, below.
Although it is obvious that the authors put a great amount of information together, the manuscript miss a good structure: the abstract seems an extension of the introduction; the introduction is quite long and it focus too much on information on the structure of the hesperidin, while lacking to provide a good rationale for the study. There is no methods section (?). I may be missing something, but the authors do not seem to follow the instructions defined by Nutrients on how to prepare a review.
Answer (A): This manuscript is a literature review, not a systematic or meta-analysis review, without Methods or Results sections, and therefore this manuscript does not follow the PRISMA guidelines.
In fact, several literature reviews are published in the Nutrients journal, following a similar structure. Just to have as example:
Nutrients 2020, 12(5), 1265; https://doi.org/10.3390/nu12051265
Nutrients 2020, 12(5), 1248; https://doi.org/10.3390/nu12051248
Abstract
The introduction of the abstract is rather long, and the authors present no results in the abstract and seem to finalize with their hypotheses. Please follow the basic structure of a study in your abstract: “introduction, methods, results, conclusion”. Even if these words are not explicit written in the abstract, the information should be there.
A: This question is in consonance with the previous answer. This manuscript is a literature review and consequently, there are not Methods or Results section neither in the Abstract nor in the body of the manuscript.
However, we have introduced a minor change to reduce the word count to less than 200 words to be in accordance with the Nutrients “Instruction for Authors”.
Introduction
The introduction is rather long. The authors could try to cut non-essential information to keep it shorter.
A: We appreciate the reviewer's constructive comment. The Introduction section has been extensively revised and non-essential information has been deleted to make the manuscript more understandable for the readers.
Line 34: “currently in the world”?
A: Thank you for your comment. As the reviewer indicated, there was a mistake in the sentence that has been corrected “Cardiovascular diseases (CVDs) are the first cause of death in the world, causing about 31% of all deaths worldwide [1].”
I suggest the authors to have a look on the references below, it may be useful for their introduction:
Collaborators GBDD. Health effects of dietary risks in 195 countries, 1990-2017: a systematic analysis for the Global Burden of Disease Study 2017. Lancet. 2019. doi:10.1016/S0140-6736(19)30041-8.
WHO - World Health Organization, editor. Vienna Declaration on Nutrition and Noncommunicable Diseases in the Context of Health 2020. WHO Ministerial Conference on Nutrition and Noncommunicable Diseases in the Context of Health 2020; 2013 4-5 July Vienna, Austria.
A: We appreciate the reviewer's constructive comment. As the reviewer proposed, it has been included the WHO reference because it is very useful in the new manuscript version, which increases the impact of the revision and they have been included in the “Introduction” section.
Line 40: in which context is diet considered an environmental factor rather than an individual factor?
A: The reviewer comment is correct. There are several evidences that non-healthy diets rich in saturated fatty acids are strongly associated with an increased risk of cardiovascular disease. Possibly this sentence can lead to misunderstandings. In the new version of the manuscript, this sentence has changed to (line 37):
“Diet is a major external factor for CVDs development…”
Additionally, to reaffirm what we want to demonstrate, we have added information from a 2018 review based on systematic reviews and meta-analysis that relates specific dietary patterns to their corresponding cardio-metabolic effects. In the text, it could be found as (line 39):
“In this sense, differences between dietary patterns, such as Mediterranean, Portfolio, Nordic and vegetarian diet are associated with different cardio-metabolic outcomes”.
Lines 42 - 45: Please be more specific about what you consider treatment and prevention. For instance, by ‘common therapies for patients with CDV disorders’, do you mean treatment for hypertension based on drugs? And these ‘alternative treatments’, are they prevention strategies for CDV? Treatment of CDV? Or hypertension? The history line is not easy to follow in those sentences, perhaps use example to better explain.
A: We want to thank the reviewer’s suggestion. We have rewritten the paragraph, improving the text adding some examples, which makes it more comprehensive. In the new version of the manuscript (line 41), the paragraph has changed to:
“Nowadays, common therapies based on drugs are addressed to patients who have already been diagnosed with any cardiovascular disorder. Taking into account all the beneficial effects associated to the diet in CVDs development, the use of alternative treatments, such as natural-based products have gained importance as a preventive strategy for improving some CVDs factors, such as hypertension, diabetes, cholesterol and obesity.”
Line 61: harmless when considering the upper-bound limits of indicated maximum consumption, according to the Dietary Reference Intakes (DRI)?
A: We have changed the word "harmless" to "innocuous" to improve the final context. Moreover, polyphenols do not have DRI, as they are not accumulated in the body and, therefore are not considered toxic. Moreover, related with polyphenol intake, there is the Estimated Dietary Intake, but is related with average consumption levels, not with toxicity limits.
Lines 63 – 68: I would suggest the authors to synthesise these explanations. The information here can be cut down to three lines.
A: In the new version of the manuscript, the paragraph has been synthesised and finally you can find it as:
“Hesperidin and, far behind, naringin, represent more than 90% of the flavonoids in sweet oranges [14,15]. The highest concentrations of hesperidin are found in the solid tissues of citrus fruits, although considerable amounts are also found in their juices [16].”
Lines 69 – 74: Since you present figure 1, the information described here on the chemical structure of the hesperidin can be deleted.
A: We appreciate the reviewer's constructive comment. We delete the redundant information about the chemical structure of Hesperidin and the final paragraph changed to:
“Hesperidin and its derived intestinal metabolites are shown in Figure 1.” (line 61)
Lines 80 – 95: this paragraph also brings too much information, please stick to the essential information that will help the reader to understand the motivation for your study.
A: We thank to the reviewer for their comments and suggestions. In this point, we would like to remark that we reduce the extension of this paragraph, removing some parts that are not relevant for the final purpose of the manuscript. In that sense, in the new manuscript version, the final paragraph is (line 72-80):
“Molecular structure of hesperidin also affects its bioavailability and absorption levels [22]. Thus, the metabolism of citrus flavanones is determined by the sugar moieties and its removal degree by intestinal bacteria. Citrus flavanones are resistant to stomach and small intestine enzymes and, because of that, reach the colon intact. There, the intestinal microbiota activity produces a breakdown in the hesperidin molecule, releasing the aglycone form named hesperitin [23,24] (Figure 1). Once inside the intestinal epithelium, hesperitin is released into the bloodstream in form of glucuronide and sulfatate conjugates [27]. In addition, an important part of the metabolized hesperitin is transformed by the microbiota present in the colon, generating some bioavailable and highly specific catabolites of hesperitin [28].”
In the abstract, which now seems like an extension of their introduction, the authors say that hesperidin has been shown to potentially contribute to obesity and CVD prevention in models, but inconsistent or null information on a protective effect of hesperidin was available for humans.
However, to my surprise, there is no information about that in the introduction, which to my understanding is the very rationale for conducting a review.
I urge the authors to substantially reduce the information on the chemical mechanisms and structure of hesperidin and improve their rationale on why this review is needed. What is the findings from studies in animal models? What kind of information is available from human studies? Why is it important to review the literature on this subject?
A: We agree with the reviewer’s comment and we have considerably reduced the introduction information about hesperidin structure and chemical mechanisms and have added more information about the rationale for the aim of the manuscript, including several sentences throughout the introduction section.
Methods? I do not understand the structure of this manuscript. I am reviewing this manuscript assuming this is a systematic review. I checked the Nutrients’ instructions for authors (https://www.mdpi.com/journal/nutrients/instructions#submission), and noticed they recommend review articles to follow the PRISMA guidelines http://prisma-statement.org/documents/PRISMA%202009%20checklist.pdf, which this manuscript does not follow. There is for instance no information about the database that the authors searched, what were the terms of the search, there is not even a Methods section.
A: This point is in consonance with the first and second questions. As pointed above, this manuscript is a literature review and consequently, not a systematic or a meta-analysis review.
Therefore, this manuscript does not follow the PRISMA guidelines, and its structure is very similar to other literature reviews are published in the journal Nutrients.
